# Optimization of Inter-group Criteria for Clustering with Minimum Size Constraints

**Eduardo S. Laber**
Department of Computer Science
PUC-Rio
Rio de Janeiro, RJ - Brazil
laber@inf.puc-rio.br

**Lucas Murtinho**
Department of Computer Science
PUC-Rio
Rio de Janeiro, RJ - Brazil
lmurtinho@aluno.puc-rio.br

## Abstract

Internal measures that are used to assess the quality of a clustering usually take into account intra-group and/or inter-group criteria. There are many papers in the literature that propose algorithms with provable approximation guarantees for optimizing the former. However, the optimization of inter-group criteria is much less understood.

Here, we contribute to the state-of-the-art of this literature by devising algorithms with provable guarantees for the maximization of two natural inter-group criteria, namely the minimum spacing and the minimum spanning tree spacing. The former is the minimum distance between points in different groups while the latter captures separability through the cost of the minimum spanning tree that connects all groups. We obtain results for both the unrestricted case, in which no constraint on the clusters is imposed, and for the constrained case where each group is required to have a minimum number of points. Our constraint is motivated by the fact that the popular `single-linkage`, which optimizes both criteria in the unrestricted case, produces clusterings with many tiny groups.

To complement our work, we present an empirical study with 10 real datasets, providing evidence that our methods work very well in practical settings.

## 1   Introduction

Data clustering is a fundamental tool in machine learning that is commonly used in exploratory analysis and to reduce the computational resources required to handle large datasets. For comprehensive descriptions of different clustering methods and their applications, we refer to Jain et al. (1999); Hennig et al. (2015). In general, clustering is the problem of partitioning a set of items so that similar items are grouped together and dissimilar items are separated. Internal measures that are used to assess the quality of a clustering (e.g. Silhouette coefficient Rousseeuw (1987) and Davies-Bouldin index Davies & Bouldin (1979)) usually take into account intra-group and inter-group criteria. The former considers the cohesion of a group while the latter measures how separated the groups are.

There are many papers in the literature that propose algorithms with provable approximation guarantees for optimizing intra-group criteria. Algorithms for $k$-center, $k$-medians and $k$-means cost functions Gonzalez (1985); Charikar et al. (2002); Ahmadian et al. (2020) are some examples. However, optimizing inter-group criteria is much less understood. Here, we contribute to the state-of-the-art by considering the maximization of two natural inter-group criteria, namely the minimum spacing and the minimum spanning tree spacing.

**Our Results.** The spacing between two groups of points, formalized in Section 2, is the minimum distance between a point in the first and a point in the second group. We consider two criteria for

37th Conference on Neural Information Processing Systems (NeurIPS 2023).

capturing the inter-group distance of a clustering, the minimum spacing (`Min-Sp`) and minimum spanning tree spacing (`MST-Sp`). The former is given by the spacing of the two groups with the smallest spacing in the clustering. The latter, as the name suggests, measures the separability of a clustering according to the cost of the minimum spanning tree (MST) that connects its groups; the largest the cost, the most separated the clustering is.

We first show that `single-linkage`, a procedure for building hierarchical clustering, produces a clustering that maximizes the MST spacing. This result contributes to a better understanding of this very popular algorithm since the only provable guarantee that we are aware of (in terms of optimizing some natural cost function) is that it maximizes the minimum spacing of a clustering Kleinberg & Tardos (2006)[Chap 4.7]. Our guarantee [Theorem 3.3] is stronger than the existing one in the sense that any clustering that maximizes the MST spacing also maximizes the minimum spacing. Figure 1 shows an example where the minimum spacing criterion does not characterize well the behavior of `single-linkage`.

Despite its nice properties, `single-linkage` tends to build clusterings with very small groups, which may be undesirable in practice. This can be seen in our experiments (see Figure 2) and is related to the well-documented fact that `single-linkage` suffers from the chaining effect (Jain et al., 1999)[Chap 3.2].

To mitigate this problem we consider the optimization of the aforementioned criteria under size constraints. Let $L$ be a given positive integer that determines the minimum size that every group in a clustering should have. A $(k, L)-$clustering is a clustering with $k$ groups in which the smallest group has at least $L$ elements. For the `Min-Sp` criterion we devise an algorithm that builds clustering with at least $L(1 - \epsilon)$ points per group while guaranteeing that its minimum spacing is not smaller than that of an optimal $(k, L)$-clustering. This result is the best possible in the sense that, unless $P = NP$, it is not possible to obtain an approximation with subpolynomial factor for the case in which the clustering is required to satisfy the hard constraint of $L$ points per group. For the `MST-Sp` criterion we also devise an algorithm with provable guarantees. It produces a clustering whose `MST-Sp` is at most a $\log k$ factor from the optimal one and whose groups have each at least $\rho L(1 - \epsilon)/2$ points, where $\rho$ is a number in the interval $[1, 2]$ that depends on the ratio $n/kL$. We also prove that the maximization of this criterion is APX-Hard for any fixed $k$.

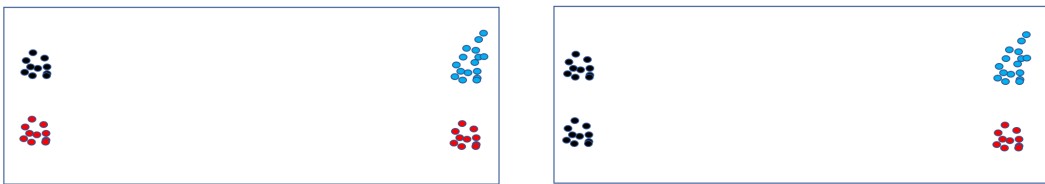

Figure 1: Partitions with 3 groups (defined by colors) that maximize the minimum spacing. The rightmost one is built by `single-linkage`, but both of them maximize the minimum spacing – showing that this condition alone is insufficient to properly characterize `single-linkage`'s behavior.

We complement our investigation with an experimental study where we compare the behaviour of the clustering produced by our proposed algorithms with those produced by $k$-means and `single-linkage` for 10 real datasets. Our algorithms, as expected, present better results than $k$-means for the criteria under consideration while avoiding the issue of small groups observed for `single-linkage`.

**Related Work.** We are only aware of a few works that propose clustering algorithms with provable guarantees for optimizing separability (inter-group) criteria. We explain some of them.

The maximum $k$-cut problem is a widely studied problem in the combinatorial optimization community and its solution can be naturally viewed as a clustering that optimizes a separability criterion. Given an edge-weighted graph $G$ and an integer $k \geq 2$, the maximum $k$-cut problem consists of finding a $k$-cut (partition of the vertices of the graph into $k$ groups) with maximum weight, where the weight of a cut is given by the sum of the weights of the edges that have vertices in different groups of the cut. The weight of the $k$-cut can be seen as a separability criterion, where the distance between two groups is given by the sum of the pairwise distances of its points. It is well-known that a random

assignment of the vertices (points) yields an $(1 - 1/k)$-approximation algorithm. This bound can be slightly improved using semi-definitive programming Frieze & Jerrum (1997).

The minimum spacing, one of the criteria we studied here, also admits an algorithm with provable guarantees. In fact, as we already mentioned, it can be maximized in polynomial time via the `single-linkage` algorithm Kleinberg & Tardos (2006)[Chap 4.7]. In what follows, we discuss works that study this algorithm as well as works that study problems and/or methods related to it.

`single-linkage` has been the subject of a number of researches (Zahn, 1971; Kleinberg, 2002; Carlsson & Mémoli, 2010; Hofmeyr, 2020). Kleinberg (2002) presents an axiomatic study of clustering, the main result of which is a proof that it is not possible to define clustering functions that simultaneously satisfy three desirable properties introduced in the paper. However, it was shown that by choosing a proper stopping rule for the `single-linkage` it satisfies any two of the three properties. Carlsson & Mémoli (2010) replaces the property of Kleinberg (2002) that the clustering function must output a partition with the property that it must generate a tree (dendogram). Then, it was established that `single-linkage` satisfies the new set of properties. In more recent work, Hofmeyr (2020) establishes a connection between minimum spacing and spectral clustering. While the aforementioned works prove that `single-linkage` has important properties, in practice, it is reported that sometimes it presents poor performance due to the so-called chaining effect (Jain et al. (1999)[Chap 3.2]).

`single-linkage` belongs to the family of algorithms that are used to build Hierarchical Agglomerative Clustering (HAC). Dasgupta (2016) frames the problem of building a hierarchical clustering as a combinatorial optimization problem, where the goal is to output a hierarchy/tree that minimizes a given cost function. The paper proposes a cost function that has desirable properties and an algorithm to optimize it. This result was improved and extended by a series of papers (Roy & Pokutta, 2016; Charikar & Chatziafratis, 2017; Cohen-Addad et al., 2019). The algorithms discussed in these papers do not seem to be employed in practice. Recently, there has been some effort to analyse HAC algorithms that are popular in practice, such as the Average Link (Moseley & Wang, 2017; Cohen-Addad et al., 2019). Our investigation of `single-linkage` can be naturally connected with this line of research.

Finally, in a recent work, Ahmadi et al. (2022) studied the notion of individual preference stability (IP-stability) in which a point is IP-stable if it is closer to its group (on average) than to any other group (on average). The clustering produced by `single-linkage` presents a kind of individual preference stability in the sense that each point is closer to some point in its group than to any point in some other group.

**Potential Applications.** We discuss two cases in which the maximization of inter-group criteria via our algorithms may be relevant: ensuring data diversity when training machine learning algorithms and population diversity in candidate solutions for genetic algorithms.

When training a machine learning model, ensuring data diversity may be crucial for achieving good results (Gong et al., 2019). In situations in which all available data cannot be used for training (e.g. training in the cloud with budget constraints), it is important to have a method for selecting a diverse subset of the data, and our algorithms can be used for this: to select $n = kL$ elements, one can partition the full data set into $k$ clusters, all of them containing at least $L$ members, and then select $L$ elements from each cluster. Using `single-linkage` to create $kL$ groups and then picking up one element per group is also a possibility but it would increase the probability of an over-representation of outliers in the obtained subset, as these outliers would likely be clustered as singletons (see Figure 2 in Section 5).

Note that our algorithms can be used to create not only one but several diverse and disjoint subsets, which might be relevant to generate partitions for cross-validation or for evaluating a model's robustness. For that, each subset is obtained by picking exactly one point per group.

For genetic algorithms, maintaining diversity over the iterations is important to ensure a good exploration of the search space (Gupta & Ghafir, 2012). If all candidate solutions become too similar, the algorithm will become too dependent on mutations for improvement, as the offspring of two solutions will likely be similar to its two parents; and mutation alone may not be enough to fully explore the search space. We can apply our algorithms in a similar manner as mentioned above, by partitioning, at each iteration, the solutions into $k$ clusters of minimum size $L$ and selecting the best $L$ solutions from each cluster, according to the objective function of the underlying optimization problem, to maintain

the solution population simultaneously optimized and diverse. Using `single-linkage` could lead to several poor solutions being selected to remain in the population, in case they are clustered as singletons.

We finally note that an algorithm that optimizes intra-group measures (e.g. $k$-means or $k$-medians) would not necessarily guarantee diversity for the aforementioned applications, as points from different groups can be close to each other.

## 2 Preliminaries

Let $\mathcal{X}$ be a set of $n$ points and let $\mathtt{dist} : \mathcal{X} \times \mathcal{X} \mapsto \mathbb{R}^+$ be a distance function that maps every pair of points in $\mathcal{X}$ into a non-negative real.

Given a $k$-clustering $\mathcal{C} = (C_1, \ldots, C_k)$, we define the spacing between two distinct groups $C_i$ and $C_j$ of $\mathcal{C}$ as

$$\mathtt{spacing}(C_i, C_j) := \min_{x \in C_i, y \in C_j} \{\mathtt{dist}(x, y)\}.$$

Then, the minimum spacing of $\mathcal{C}$ is given by

$$\mathtt{Min\text{-}Sp}(\mathcal{C}) := \min_{\substack{C_i, C_j \in \mathcal{C} \\ i \neq j}} \{\mathtt{spacing}(C_i, C_j)\}$$

A $k$-clustering $\mathcal{C}$ induces on $(\mathcal{X}, \mathtt{dist})$ an edge-weighted complete graph $G_{\mathcal{C}}$ whose vertices are the groups of $\mathcal{C}$ and the weight $w(e_{ij})$ of the edge $e_{ij}$ between groups $C_i$ and $C_j$ is given by $w(e_{ij}) = \mathtt{spacing}(C_i, C_j)$. For a set of edges $S \in G_{\mathcal{C}}$ we define $w(S) := \sum_{e \in S} w(e)$.

The *minimum spanning tree spacing* of $\mathcal{C}$ ($\mathtt{MST\text{-}Sp}(\mathcal{C})$) is defined as the sum of the weights of the edges of the minimum spanning tree ($\mathrm{MST}(G_{\mathcal{C}})$) for $G_{\mathcal{C}}$. In formulae,

$$\mathtt{MST\text{-}Sp}(\mathcal{C}) := \sum_{e \in MST(G_{\mathcal{C}})} w(e).$$

Here, we will be interested in the problems of finding partitions with maximum `Min-Sp` and maximum `MST-Sp` both in the unrestricted case in which no constraint on the groups is imposed and in the constrained case where each group is required to have at least $L$ points.

**Single-Linkage**. We briefly explain `single-linkage`. The algorithm starts with $n$ groups, each of them consisting of a point in $\mathcal{X}$. Then, at each iteration, it merges the two groups with minimum spacing into a new group. Thus, by the end of the iteration $n - k$ it obtains a clustering with $k$ groups. In Kleinberg & Tardos (2006) it is proved that the `single-linkage` obtains a $k$-clustering with maximum minimum spacing.

**Theorem 2.1** (Kleinberg & Tardos (2006), chap 4.7)**.** *The* `single-linkage` *algorithm obtains the $k$-clustering with maximum* `Min-Sp` *for instance* $(\mathcal{X}, \mathtt{dist})$*.*

`single-linkage` and minimum spanning trees are closely related since the former can be seen as the Kruskal's algorithm for building MST's with an early stopping rule. To analyze our algorithms we make use of well-known properties of MST's as the cut property[Kleinberg & Tardos (2006), Property 4.17] and the cycle property[Kleinberg & Tardos (2006), Property 4.20]. Their statements can be found in the appendix.

For ease of presentation, we assume that all values of `dist` are distinct. We note, however, that our results hold if this assumption is dropped.

## 3 Relating `Min-Sp` and `MST-Sp` criteria

We show that `single-linkage` finds the clustering with maximum `MST-Sp` and the maximization of `MST-Sp` implies the maximization of `Min-Sp`. These results are a consequence of Lemma 3.1 that generalizes the result of Theorem 2.1. The proof of this lemma can be found in the appendix.

Fix an instance $I = (\mathcal{X}, \mathtt{dist}, k)$. In what follows, $\mathcal{C}_{SL}$ is a $k$-clustering obtained by `single-linkage` for instance $I$ and $T_{SL}$ is a MST for $G_{\mathcal{C}_{SL}}$. Moreover, $w_i^{SL}$ is the weight of the $i$-th smallest weight of $T_{SL}$.

**Lemma 3.1.** *Let $\mathcal{C}$ be a $k$-clustering for $I$ and let $w_i$ be the weight of the $i$-th smallest weight in a MST $T$ for the graph $G_{\mathcal{C}}$. Then, $w_i^{SL} \geq w_i$.*

**Theorem 3.2.** *The clustering $\mathcal{C}_{SL}$ returned by* `single-linkage` *for instance $(\mathcal{X}, \text{dist}, k)$ maximizes the* `MST-Sp` *criterion.*

*Proof.* Let $\mathcal{C}$ be a $k$-clustering for $(\mathcal{X}, \text{dist}, k)$ and let $w_i$ be the weight of the $i$-th cheapest edge of the MST for $G_{\mathcal{C}}$. Since $w_i^{SL} \geq w_i$ for $i = 1, \ldots, k-1$, we have that

$$\text{MST-Sp}(\mathcal{C}_{SL}) = \sum_{i=1}^{k-1} w_i^{SL} \geq \sum_{i=1}^{k-1} w_i = \text{MST-Sp}(\mathcal{C})$$

$\square$

**Theorem 3.3.** *Let $\mathcal{C}^*$ be a clustering that maximizes the* `MST-Sp` *criterion for instance $(\mathcal{X}, \text{dist}, k)$. Then, it also maximizes* `Min-Sp` *for this same instance.*

*Proof.* Let us assume that $\mathcal{C}^*$ maximizes the `MST-Sp` criterion but it does not maximize the `Min-Sp` criterion. Thus, $w_1^{SL} > w_1^*$, where $w_1^*$ is the minimum spacing of $\mathcal{C}^*$. It follows from the previous lemma that

$$\text{MST-Sp}(\mathcal{C}_{SL}) = \sum_{i=1}^{k-1} w_i^{SL} > \sum_{i=1}^{k-1} w_i^* = \text{MST-Sp}(\mathcal{C}^*),$$

which contradicts the assumption that $\mathcal{C}^*$ maximizes the `MST-Sp` criterion. $\square$

The next example (in the spirit of Figure 1) shows that a partition that maximizes the `Min-Sp` criterion may have a poor result in terms of the `MST-Sp` criterion.

**Example 3.4.** *Let $D$ be a positive number much larger than $k$. Moreover, let $[t]$ be the set of the $t$ first positive integers and $S = \{(D \cdot i, j) | i, j \in [k-1]\} \cup (D, k)$ be a set of $(k-1)^2 + 1$ points in $\mathbb{R}^2$.*

`single-linkage` *builds a $k$-clustering with* `Min-Sp` *$1$ and* `MST-Sp` $1 + (k-2)D$ *for $S$.*

*However, the $k$-clustering $(C_1, \ldots, C_k)$, where $C_j = \{(D \cdot i, j) | i = 1, \ldots, k-1\}$, for $j < k$ and $C_k = \{(D, k)\}$ has* `Min-Sp` *$1$ and* `MST-Sp` $= (k-1)$,

## 4 Avoiding small groups

In this section, we optimize our criteria under the constraint that all groups must have at least $L$ points, where $L$ is a positive integer not larger than $n/k$ that is provided by the user. Note that the problem is not feasible if $L > n/k$.

We say that an algorithm has $(\beta, \gamma)$-approximation for a criterion $\kappa \in \{\text{Min-Sp}, \text{MST-Sp}\}$ if for all instances $I$ it obtains a clustering $\mathcal{C}_I$ such that $\mathcal{C}_I$ is a $(k, \lfloor \beta L \rfloor)$-clustering and the value of $\kappa$ for $\mathcal{C}_I$ is at least $\gamma \cdot OPT$, where $OPT$ is the maximum possible value of $\kappa$ that can be achieved for a $(k, L)$-clustering.

We first show how to obtain a $(1 - \epsilon, 1)$ for the `Min-Sp` criterion.

### 4.1 The `Min-Sp` criterion

We start with the polynomial-time approximation scheme for the `Min-Sp` criterion. Our method uses, as a subroutine, an algorithm for the max-min scheduling problem with identical machines Csirik et al. (1992); Woeginger (1997). Given $m$ machines and a set of $n$ jobs, with processing times $p_1, \ldots, p_n$, the problem consists of finding an assignment of jobs to the machines so that the load of the machine with minimum load is maximized. This problem admits a polynomial-time approximation scheme Woeginger (1997).

Let `MaxMinSched`$(P, k, \epsilon)$ be a routine that implements this scheme. It receives as input a parameter $\epsilon > 0$, an integer $k$ and a list of numbers $P$ (corresponding to processing times). Then, it returns a partition of $P$ into $k$ lists (corresponding to machines) such that the sum of the numbers of the list

with minimum sum is at least $(1 - \epsilon)OPT$, where $OPT$ is the minimum load of a machine in an optimal solution of for the max-min scheduling when the list of processing times is $P$ and the number of machines is $k$.

Algorithm 1, as proved in the next theorem, obtains a $(k, L(1 - \epsilon))$-clustering whose `Min-Sp` is at least the `Min-Sp` of an optimal $(k, L)$-clustering. For that, it looks for the largest integer $t$ for which the clustering $\mathcal{A}_t$ obtained by executing $t$ steps of `single-linkage` and then combining the resulting groups into $k$ groups (via `MaxMinSched`) is a $(k, L(1 - \epsilon))$-clustering. We assume that `MaxMinSched`, in addition of returning the partition of the sizes, also returns the group associated to each size.

---

**Algorithm 1** `AlgoMinSp`$(\mathcal{X}; \texttt{dist}; k; \epsilon > 0; L)$

---

$t \leftarrow n - k$
**while** $t \geq 0$ **do**
    Run $t$ merging steps of the `single-linkage` for input $\mathcal{X}$
    Let $C_1, \ldots, C_{n-t}$ the groups obtained by the end of the $t$ steps
    $P \leftarrow (|C_1|, \ldots, |C_{n-t}|)$
    $A_t \leftarrow$ `MaxMinSched`$(P, k, \epsilon)$
    **if** the smallest group in $A_t$ has size greater than or equal to $L(1 - \epsilon)$ **then**
        Return $A_t$
    **else**
        $t \leftarrow t - 1$

---

**Theorem 4.1.** *Fix $\epsilon > 0$. The clustering $\mathcal{A}_t$ returned by the Algorithm 1 is a $(k, (1 - \epsilon)L)$-clustering that satisfies Min-Sp$(\mathcal{A}_t) \geq$ Min-Sp$(\mathcal{C}^*)$, where $\mathcal{C}^*$ is the $(k, L)$-clustering with maximum* `Min-Sp`.

*Proof.* By design, $\mathcal{A}_t$ has $k$ groups with at least $L(1 - \epsilon)$ points in each of them.

For the sake of contradiction, let us assume that Min-Sp$(\mathcal{A}_t) <$ Min-Sp$(\mathcal{C}^*)$.

Let $\mathcal{C} = (C_1, \ldots, C_{n-t})$ be the list of $n - t$ groups obtained when $t$ merging steps of `single-linkage` are performed. We assume w.l.o.g. that $C_1$ and $C_2$ are the two groups with a minimum spacing in this list, so that Min-Sp$(\mathcal{C}) =$ `spacing`$(C_1, C_2)$. Since $\mathcal{A}_t$ is a $k$-clustering that is obtained by merging groups in $\mathcal{C}$ we have Min-Sp$(\mathcal{A}_t) \geq$ Min-Sp$(\mathcal{C}) =$ `spacing`$(C_1, C_2)$.

For $i = 1, \ldots, n-t$ we have that $C_i \subseteq C$ for some $C \in \mathcal{C}^*$, otherwise we would have Min-Sp$(\mathcal{A}_t) \geq$ Min-Sp$(\mathcal{C}^*)$. In addition, we must have $C_1 \cup C_2 \subseteq C$ for some $C \in \mathcal{C}^*$, otherwise, again, we would have Min-Sp$(\mathcal{A}_t) \geq$ Min-Sp$(\mathcal{C}^*)$.

We can conclude that there is a feasible solution with minimum load not smaller than $L$ for the max-min scheduling problem with processing times $P' = (|C_1 \cup C_2|, |C_3|, \ldots, |C_{n-t}|)$ and $K$ machines. Thus, by running $t + 1$ steps of `single-linkage` followed by `MaxMinSched`$(P', k, \epsilon)$, we would get a $k$-clustering whose smallest group has at least $L(1 - \epsilon)$ points. This implies that the algorithm would have stopped after performing $t + 1$ merging steps, which is a contradiction. $\square$

---

Algorithm 1, as presented, may run `single-linkage` $n - k$ times, which may be quite expensive. Fortunately, it admits an implementation that runs `single-linkage` just once, and performs an inexpensive binary search to find a suitable $t$.

The next theorem shows that Algorithm 1 has essentially tight guarantees under the hypothesis that $P \neq NP$. The proof can be found in the appendix

**Theorem 4.2.** *Unless $P = NP$, for any $\alpha = poly(n)$, the problem of finding the $(k, L)$-clustering that maximizes the* `Min-Sp` *criterion does not admit a $(1, \frac{1}{\alpha})$-approximation.*

## 4.2 The `MST-Sp` criterion

Now, we turn to the `MST-Sp` criterion. Let $\rho := \min\{\frac{n/k}{L}, 2\}$. Our main contribution is Algorithm 2, it obtains a $(\frac{\rho(1-\epsilon)}{2}, \frac{1}{H_{k-1}})$ approximation for this criterion, where $H_{k-1} = \sum_{i=1}^{k-1} \frac{1}{i}$ is the $(k-1)$-th Harmonic number. Note that $H_{k-1}$ is $\Theta(\log k)$.

In high level, for each $\ell = 2, \ldots, k$, the algorithm calls `AlgoMinSp` (Algorithm 1) to build a clustering $\mathcal{A}'_\ell$ with $\ell$ groups and then it transforms $\mathcal{A}'_\ell$ (lines 5-13) into a clustering $\mathcal{A}_\ell$ with $k$ groups. In the end, it returns the clustering, among the $k - 1$ considered, with maximum `MST-Sp`.

---

**Algorithm 2** `Constrained-MaxMST`$(\mathcal{X}; \mathtt{dist}; k; L; \epsilon)$

---

1: **for** $\ell = 2, \ldots, k$ **do**
2:     $\mathcal{A}'_\ell \leftarrow$ `AlgoMinSp`$(\mathcal{X},\mathtt{dist},\ell,\epsilon,L)$
3:     `NonVisited` $\leftarrow \mathcal{A}'_\ell$
4:     $\mathcal{A}_\ell \leftarrow \emptyset$
5:     **for** each $A'$ in $\mathcal{A}'_\ell$, iterating from the largest group to the smallest **do**
6:        `NonVisited` $\leftarrow$ `NonVisited` $- A'$
7:        `SplitNumber` $\leftarrow \left\lfloor \frac{2|A'|}{\rho(1-\epsilon)L} \right\rfloor$
8:        **if** $|\mathcal{A}_\ell| + |$ `NonVisited` $| +$ `SplitNumber` $< k$ **then**
9:           Split $A'$ into `SplitNumber` as balanced as possible groups and add them to $\mathcal{A}_\ell$
10:        **else**
11:           Split $A'$ into $k - |\mathcal{A}_\ell| - |$ `NonVisited` $|$ as balanced as possible groups; add them to $\mathcal{A}_\ell$
12:           Add all groups in `NonVisited` to $\mathcal{A}_\ell$
13:           Break
14: **Return** the clustering $A_\ell$, among the $k - 1$ obtained, that has the maximum `MST-Sp`

---

We remark that we do not need to scan the groups in $\mathcal{A}'_\ell$ by non-increasing order of their sizes to establish our guarantees presented below. However, this rule tends to avoid groups with sizes smaller than $L$.

**Lemma 4.3.** *Fix $\epsilon > 0$. Thus, for each $\ell$, every group in $\mathcal{A}_\ell$ has at least $\left\lfloor \frac{\rho(1-\epsilon)L}{2} \right\rfloor$ points.*

*Proof.* The groups that are added to $\mathcal{A}_\ell$ in line 12 have at least $(1 - \epsilon)L$ points while the number of points of those that are added at either line 11 or 9 is at least

$$\left\lfloor \frac{|A'|}{\lfloor 2|A'|/\rho(1-\epsilon)L \rfloor} \right\rfloor \geq \left\lfloor \frac{\rho(1-\epsilon)L}{2} \right\rfloor$$

Moreover, if the **For** is not interrupted by the **Break** command, the total number of groups in $\mathcal{A}_\ell$ is

$$\sum_{A' \in \mathcal{A}'_\ell} \left\lfloor \frac{2|A'|}{\rho(1-\epsilon)L} \right\rfloor \geq \sum_{A' \in \mathcal{A}'_\ell} \frac{2|A'|}{\rho(1-\epsilon)L} - \ell \geq \frac{2n}{\rho(1-\epsilon)L} - k \geq \frac{2k}{(1-\epsilon)} - k \geq k$$

Since the **For** is interrupted as soon as $k$ groups can be obtained then, $\mathcal{A}_\ell$ has $k$ groups. $\qquad \square$

For the next results, we use $\mathcal{C}^*$ to denote the $(k, L)$-clustering with maximum `MST-Sp` and $w_i^*$ to denote the cost of the $i$-th cheapest edge in the MST for $G_{\mathcal{C}^*}$. Our first lemma can be seen as a generalization of Theorem 4.1. Its proof can be found in the appendix.

**Lemma 4.4.** *For each $\ell$, `Min-Sp`$(\mathcal{A}'_\ell) \geq w^*_{k-\ell+1}$.*

A simple consequence of the previous lemma is that the `MST-Sp` of clustering $\mathcal{A}'_\ell$ is at least $(\ell - 1) \cdot w^*_{k-\ell+1}$. The next lemma shows that this bound also holds for the clustering $\mathcal{A}_\ell$. The proof consists of showing that each edge of a MST for $\mathcal{A}'_\ell$ is also an edge of a MST for $\mathcal{A}_\ell$.

**Lemma 4.5.** *For each $\ell = 2, \ldots, k$ we have `MST-Sp`$(\mathcal{A}_\ell) \geq (\ell - 1) \cdot w^*_{k-\ell+1}$.*

*Proof.* Let $T_\ell$ and $T'_\ell$ be, respectively, the MST for $G_{\mathcal{A}_\ell}$ and $G_{\mathcal{A}'_\ell}$. By the previous lemma, each of the $(\ell - 1)$ edges of $T'_\ell$ has cost at least `Min-Sp`$(\mathcal{A}'_\ell) \geq w^*_{k-\ell+1}$. Thus, to establish the result, it is enough to argue that each edge of $T'_\ell$ also belongs to $T_\ell$.

We say that a group $A \in \mathcal{A}_\ell$ is is generated from a group $A' \in \mathcal{A}'_\ell$ if $A = A'$ or $A$ is one of the balanced groups that is generated when $A'$ is split in the internal **For** of Algorithm 2. We say that a

vertex $x$ in $G_{\mathcal{A}_\ell}$ is generated from a vertex $x'$ in $G_{\mathcal{A}'_\ell}$ if the group corresponding to $x$ is generated by the corresponding to $x'$.

Let $e' = u'v'$ be an edge in $T'_\ell$ and let $S'$ be a cut in graph $G_{\mathcal{A}'_\ell}$ whose vertices are those from the connected component of $T'_\ell \setminus e'$ that includes $u'$. We define the cut $S$ of $G_{\mathcal{A}_\ell}$ as follows $S = \{x \in G_{\mathcal{A}_\ell} | x$ is generated from some $x' \in S'\}$.

Let $u$ and $v$ be vertices generated from $u'$ and $v'$, respectively, that satisfy $w(uv) = w(u'v')$. It is enough to show that $uv$ is the cheapest edge that crosses $S$ since by the cut property [Kleinberg & Tardos (2006), Property 4.17.] this implies that $uv \in T_\ell$. We prove it by contradiction. Let us assume that there is another edge $f = yz$ that crosses $S$ and has weight smaller than $w(uv)$. Let $y'$ and $z'$ be vertices in $G_{\mathcal{A}'_\ell}$ that generate $y$ and $z$, respectively, and let $f' = y'z'$. Thus, $w(f') \leq w(f) < w(uv) = w(u'v') = w(e')$. However, this contradicts the cycle property of MST's [Kleinberg & Tardos (2006), Property 4.20] because it implies that the edge with the largest weight in the cycle of $G_{\mathcal{A}'_\ell}$ comprised by edge $f'$ and the path in $T'_\ell$ the connects $y'$ to $z'$ belongs to the $T'_\ell$. $\quad\square$

The next theorem is the main result of this section.

**Theorem 4.6.** *Fix $\epsilon > 0$. Algorithm 2 is a $\left(\frac{(1-\epsilon)\rho}{2}, \frac{1}{H_{k-1}}\right)$-approximation for the problem of finding the $(k, L)$–clustering that maximizes the* MST-Sp *criterion.*

*Proof.* Let $\mathcal{C}$ be the clustering returned by Algorithm 2. Lemma 4.3 guarantees that $\mathcal{C}$ is a $(k, \lfloor \frac{(1-\epsilon)\rho L}{2} \rfloor)$-clustering.

Thus, we just need to argue about MST-Sp$(\mathcal{C})$. We have that

$$\text{MST-Sp}(\mathcal{C}^*) = \sum_{i=2}^{k} w^*_{k-i+1}$$

and, due to Lemma 4.5, MST-Sp$(\mathcal{C}) \geq \max\{(\ell - 1) \cdot w^*_{k-\ell+1} | 2 \leq \ell \leq k\}$.

Let $\tilde{\ell}$ be the value $\ell$ that maximizes $(\ell-1) \cdot w^*_{k-\ell+1}$. It follows that $w^*_{k-i+1} \leq ((\tilde{\ell}-1)/(i-1)) w^*_{k-\tilde{\ell}+1}$. for $i = 2, \ldots, k$. Thus,

$$\frac{\text{MST-Sp}(\mathcal{C}^*)}{\text{MST-Sp}(\mathcal{C})} \leq \frac{\sum_{i=2}^{k} w^*_{k-i+1}}{(\tilde{\ell}-1) \cdot w^*_{k-\tilde{\ell}+1}} \leq \frac{(\tilde{\ell}-1) \cdot w^*_{k-\tilde{\ell}+1} \cdot \sum_{i=2}^{k} \frac{1}{i-1}}{(\tilde{\ell}-1) \cdot w^*_{k-\tilde{\ell}+1}} = H_{k-1}$$

$\square$

We end this section by showing that the optimization of MST-Sp is APX-HARD (for fixed $k$) when a hard constraint on the number of points per group is imposed. The proof can be found in the appendix.

**Theorem 4.7.** *Unless $P = NP$, for any $\alpha = poly(n)$, there is no $(1, \frac{k-2}{k-1} + \frac{1}{\alpha(k-1)})$-approximation for the problem of finding the $(k, L)$-clustering that maximizes the* MST-Sp *criterion.*

## 5 Experiments

To evaluate the performance of Algorithms 1 and 2, we ran experiments with 10 different datasets, comparing the results with those of single-linkage and of the traditional $k$-means algorithm from Lloyd (1982) with a ++ initialization (Arthur & Vassilvitskii, 2007). For the implementation of routine MaxMinSched, employed by Algorithm 1, we used the Longest Processing Time rule. This rule has the advantage of being fast while guaranteeing a $3/4$ approximation for the max-min scheduling problem (Csirik et al., 1992). The code for running the algorithms can be found at https://github.com/lmurtinho/SizeConstrainedSpacing.

Our first experiment investigates the size of the groups produced by single-linkage for the 10 datasets, whose dimensions can be found in the first two columns of Table 1. Figure 2 shows the proportion of singletons for each dataset with the growth of $k$. For all datasets but Vowel and Mice

Table 1: `Min-Sp` and `MST-Sp` for the different methods and datasets.

| | Dimensions | | Min-Sp | | | MST-Sp | | |
| | n | k | Algo 1 | Algo 2 | k-means | Algo 1 | Algo 2 | k-means |
|---|---|---|---|---|---|---|---|---|
| anuran | 7,195 | 10 | **0.19** | 0.09 | 0.05 | 1.71 | **1.87** | 1.01 |
| avila | 20,867 | 12 | **0.07** | 0.04 | 0 | 0.77 | **0.81** | 0.66 |
| collins | 1,000 | 30 | **0.42** | **0.42** | 0.22 | **12.42** | **12.42** | 8.58 |
| digits | 1,797 | 10 | **19.74** | **19.74** | 13.79 | **178.22** | **178.22** | 145.13 |
| letter | 20,000 | 26 | **0.2** | 0.11 | 0.07 | 4.98 | **5.67** | 1.98 |
| mice | 552 | 8 | **0.79** | **0.79** | 0.24 | **5.66** | **5.66** | 2.37 |
| newsgroups | 18,846 | 20 | **1** | 1 | 0.17 | 19 | **19** | 8.4 |
| pendigits | 10,992 | 10 | **23.89** | 9.08 | 8.31 | 215.11 | **217.01** | 119.85 |
| sensorless | 58,509 | 11 | **0.13** | 0.08 | 0.03 | 1.31 | **1.36** | 1.29 |
| vowel | 990 | 11 | **0.49** | **0.49** | 0.11 | **4.94** | **4.94** | 1.84 |

the majority of groups are singletons, even for small values of $k$. This undesirable behavior motivates our constraint on the minimum size of a group.

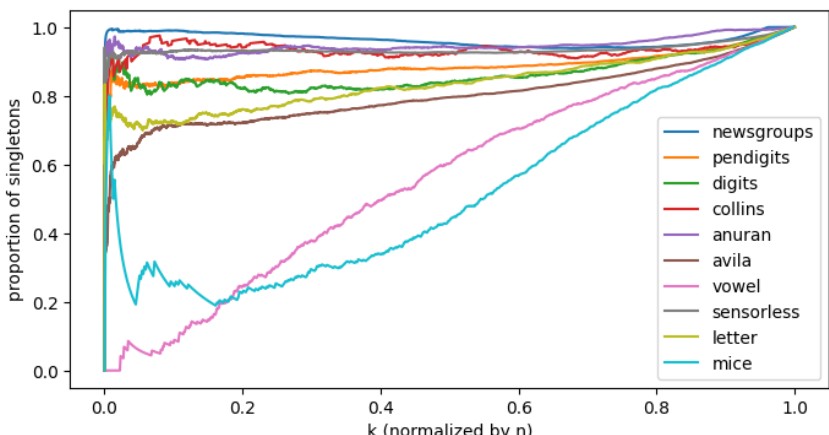

Figure 2: Proportion of singletons for each dataset with the growth of $k$

In our second experiment, we compare the values of `Min-Sp` and `MST-Sp` achieved by our algorithms with those of $k$-means. While $k$-means is not a particularly strong competitor in the sense that it was not designed to optimize our criteria, the motivation to include it is that $k$-means is a very popular choice among practitioners. Moreover, for datasets with well-separated groups, the minimization of the squared sum of errors (pursued by $k$-means) should also imply the maximization of inter-group criteria.

Table 1 presents the results of this experiment. The values chosen for $k$ are the numbers of classes (dependent variable) in the datasets, while for the `dist` function we employed the Euclidean distance. The values associated with the criteria are averages of 10 executions, with each execution corresponding to a different seed provided to $k$-means. To set the value of $L$ for the $i$-th execution of our algorithms, we take the size of the smallest group generated by $k$-means for this execution and multiply it by $4/3$. This way, we guarantee that the size of the smallest group produced by our methods, for each execution, is not smaller than that of $k$-means, which makes the comparison among the optimization criteria fairer.

With respect to the `Min-Sp` criterion, Algorithm 1 is at least as good as Algorithm 2 for every dataset (being superior on 6) and both Algorithm 1 and 2 outperform $k$-means on all datasets. On the other hand, with respect to the `MST-Sp` criterion, Algorithm 2 is at least as good as Algorithm 1 for every dataset (being better on 6) and, again, both algorithms outperform $k$-means for all datasets. In the appendix, we present additional information regarding this experiment. In particular, we show that

Table 2: Running time in seconds of `single-linkage` and our methods.

| Dataset | single-linkage | Algo 1 | Algo 2 |
|---|---|---|---|
| sensorless | 93.8 | 99.4 | 701.4 |
| newsgroups | 274.2 | 276.4 | 440.4 |
| letter | 4.6 | 5.9 | 116.4 |
| avila | 4.0 | 5.5 | 62.5 |
| pendigits | 1.4 | 2.1 | 16.0 |

the `MST-Sp` achieved by Algorithm 2 is on average $81\%$ of the upper bound $\sum_{\ell=2}^{k} \text{Min-Sp}(\mathcal{A}'_{\ell})$ that follows from Lemma 4.4. This is much better than the $1/H_{k-1}$ ratio given by the theoretical bound.

Finally, Table 2 shows the running time of our algorithms for the datasets that consumed more time. We observe that the overhead introduced by Algorithm 1 w.r.t. `single-linkage` is negligible while Algorithm 2, as expected, is more costly. In the appendix, we show that a strategy that only considers values of $\ell$ that can be written as $\lceil k/2^t \rceil$, for $t = 0, \ldots, \lfloor \log k \rfloor$, in the first loop of Algorithm 2 provides a significant gain of running time while incurring a small loss in the `MST-Sp`. We note that the $\log k$ bound of Theorem 4.6 is still valid for this strategy.

## 6 Final Remarks

We have endeavored in this paper to expand the current knowledge on clustering methods for optimizing inter-cluster criteria. We have proved that the well-known `single-linkage` produces partitions that maximize not only the minimum spacing between any two clusters, but also the MST spacing, a stronger guarantee. We have also studied the task of maximizing these criteria under the constraint that each group of the clustering has at least $L$ points. We provided complexity results and algorithms with provable approximation guarantees.

One potential limitation of our proposed algorithms is their usage on massive datasets (in particular Algorithm 2) since they execute `single-linkage` one or many times. If the `dist` function is explicitly given, then the $\Omega(n^2)$ time spent by `single-linkage` is unavoidable. However, if the distances can be calculated from the set of points $\mathcal{X}$ then faster algorithms might be obtained.

The main theoretical question that remained open in our work is whether there exist constant approximation algorithms for the maximization of `MST-Sp`. In addition to addressing this question, interesting directions for future research include handling different inter-group measures as well as other constraints on the structure of clustering.

## Acknowledgments and Disclosure of Funding

The authors thank BigDataCorp (`https://bigdatacorp.com.br/`) for providing computational power for the experiments of an initial version of the paper.

The work of the authors is partially supported by the Air Force Office of Scientific Research (award number FA9550-22-1-0475).

The work of the first author is partially supported by CNPq (grant 310741/2021-1).

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

## A  Properties of Minimum Spanning Trees

**Theorem A.1** (Cut Property, Kleinberg & Tardos (2006), Property 4.17.)**.** *Let $G = (V, E)$ be a graph with distinct weights on its edges. Let $S \subset V$ be a non-empty cut in $G$, If $e$ is the edge with minimum cost among those that have one endpoint in $S$ and the other one in $V \setminus S$, then $e$ belongs to every MST for $G$*

**Theorem A.2** (Cycle Property, Kleinberg & Tardos (2006), Property 4.20.)**.** *Let $G = (V, E)$ be a graph with distinct weights on its edges and let $C$ be a cycle in $G$. Then, the edge with the largest weight in $C$ does not belong to any minimum spanning tree for $G$.*

The following characterization of MST's will be useful. Its correctness follows directly from the cycle property.

**Theorem A.3.** *Let $G = (V, E)$ be a graph with weights on its edges. A spanning tree $T$ for $G$ is a MST for $G$ if and only if for each edge $e = uv$ in $E$, the weight $w(e)$ of $e$ satisfies $w(e) \geq w(e')$ for every edge $e'$ in the path that connects $u$ to $v$ in $T$.*

## B  Proof of Lemma 3.1

We need the following proposition.

**Proposition B.1.** *Let $\mathcal{C}'$ be a $k$-clustering for instance $I$ and let $T'$ be a MST for $G_{\mathcal{C}'}$. Moreover, let $C_i'$ and $C_j'$ be groups of $\mathcal{C}'$ such that $\mathtt{spacing}(C_i', C_j') = \mathtt{Min\text{-}Sp}(\mathcal{C}')$. Then, the tree $T^a$ that results from the contraction of the nodes $C_i'$ and $C_j'$ in $T'$ is a MST for $G_{\mathcal{C}^a}$, where $\mathcal{C}^a$ is the $(k-1)$-clustering obtained from $\mathcal{C}'$ by merging $C_i'$ and $C_j'$.*

*Proof.* We show that $T^a$ satisfies the conditions of Theorem A.3 when $G = G_{\mathcal{C}^a}$. For that, we will use the fact that $T'$ satisfies the conditions of Theorem A.3 when $G = G_{\mathcal{C}'}$.

Let $x$ and $y$ be nodes of $T^a$. For the sake of contradiction, we assume that edge $xy$ does not satisfy the conditions of Theorem A.3 when $G = G_{\mathcal{C}^a}$ and $T = T^a$. Let $w^*$ be the weight of edge $xy$ and let $e$ be an edge in the path that connects $x$ to $y$ in $T^a$ such that $w(e) > w^*$. We have two cases:

Case 1) $x \neq C_i' \cup C_j'$ and $y \neq C_i' \cup C_j'$. Then, $e$ is also an edge in the path that connects $x$ to $y$ in $T'$. This implies that $xy$ does not satisfy the required conditions when $G = G_{\mathcal{C}'}$ and $T = T'$, which is a contradiction.

Case 2) $x = C_i' \cup C_j'$ or $y \neq C_i' \cup C_j'$. Let us assume w.l.o.g. that $x = C_i' \cup C_j'$. Let $w_i'$ and $w_j'$ be, respectively, the weights of the edges $(y, C_i')$ and $(y, C_j')$ in $G_{\mathcal{C}'}$. We have that $w^* = \min\{w_i', w_j'\}$.

Let us assume w.l.o.g. that $w^* = w_i'$. Then, $e$ is also an edge in the path that connects $y$ to $C_i'$ in $T'$. Again, this implies that $xy$ does not satisfy the required conditions when $G = G_{\mathcal{C}'}$ and $T = T'$, which is a contradiction. □

*Proof of Lemma 3.1.*

It follows from the previous proposition hat the tree $T^{i-1}$ that is obtained by contracting the $i-1$ cheapest edges of $T$ is a MST for $G_{\mathcal{C}^{i-1}}$, where $\mathcal{C}^{i-1}$ is a clustering for instance $I$ that contains $(k - (i-1))$ groups. The cheapest edge of $T^{i-1}$ is exactly the $i$-th cheapest edge of $T$. Thus, $\mathtt{Min\text{-}Sp}(\mathcal{C}^{i-1}) = w_i$.

Similarly, $w_i^{SL}$ is exactly the minimum spacing of a clustering, with $(k - (i-1))$ groups, that is obtained by $\mathtt{single\text{-}linkage}$ for instance $I$. Thus, it follows from Theorem 2.1 that $w_i^{SL} \geq w_i$.

## C  Proofs of Section 4

### C.1  Proof of Lemma 4.4

*Proof.* Let $T^*$ be the MST for $G_{\mathcal{C}^*}$. If we remove the $\ell - 1$ most expensive edges of $T^*$ we obtain a forest $F$ with $\ell$ connected components. The clustering $\mathcal{C}_\ell^*$ comprised by $\ell$ groups in

which the $i$th group corresponds to the $i$th connected component of $F$ is a $(\ell, L)$-clustering and Min-Sp$(\mathcal{C}_\ell^*) = w_{k-\ell+1}^*$.

Let $OPT$ be the Min-Sp of the $(\ell, L)$-clustering with maximum Min-Sp. Thus, by Theorem 4.1
$$\text{Min-Sp}(\mathcal{A}_\ell') \geq OPT \geq \text{Min-Sp}(\mathcal{C}_\ell^*) = w_{k-\ell+1}^* \qquad \square$$

## C.2   Proof of Theorem 4.2

*Proof.* We make a reduction from the $(T/4, T/2)$-restricted 3-PARTITION problem. Given a multiset $S = \{s_1, \ldots, s_{3m}\}$ of positive integers that satisfies $\sum_i s_i = mT$, the 3-PARTITION problem consists of deciding whether or not there exists a partition of $S$ into $m$ triples such that the sum of the numbers in each one is equal to $T$. In the $(T/4, T/2)$-restricted 3-PARTITION problem, there is an additional requirement that each number of $S$ should be in the interval $(T/4, T/2)$. This problem is strongly NP-COMPLETE Garey & Johnson (1979)

The instance $I = (X, k, L, \texttt{dist})$ for our clustering problem is built as follows: we set $L = T$, $k = m$; for $i = 1, \ldots, 3m$ let $\mathcal{X}_i$ be a set with $s_i$ points so that the distance between points in the same group $\mathcal{X}_i$ is 1 while the distance between points in different groups is $\alpha + 1$. We set $\mathcal{X} = \mathcal{X}_1 \cup \ldots \cup \mathcal{X}_{3m}$. Note that we are employing a pseudo-polynomial reduction but this is fine since the $(T/4, T/2)$-restricted 3-PARTITION problem is strongly NP-COMPLETE.

Let us assume that there is an $(1, 1/\alpha)$-approximation for our problem and let $\mathcal{C}$ be the clustering returned by this algorithm for instance $I$. We argue that the answer to the 3-PARTITION problem is 'YES' if and only if Min-Sp$(\mathcal{C}) = \alpha + 1$.

First, we show that if the answer is 'YES', there is a $k$-clustering $\mathcal{C}^*$ for $I$ with Min-Sp$(\mathcal{C}^*) = \alpha + 1$. In fact, let $S_1, \ldots, S_m$ be a solution of the 3-PARTITION problem and let $\{s_{i_1}, s_{i_2}, s_{i_3}\}$ the numbers in $S_i$. Let $\mathcal{C}^*$ be a $k$-clustering where the $i$th group is comprised by all points in $\mathcal{X}_{i_1} \cup \mathcal{X}_{i_2} \cup \mathcal{X}_{i_3}$. Clearly, each group has $L$ points and the Min-Sp of this clustering is $\alpha + 1$. Since our algorithm is a $(1, 1/\alpha)$-approximation it returns a clustering $\mathcal{C}$ with Min-Sp$(\mathcal{C}) \geq (\alpha + 1)/\alpha$. Since the Min-Sp of any clustering for instance $I$ is either 1 or $\alpha + 1$ we have that Min-Sp$(\mathcal{C}) = \alpha + 1$

On the other hand, if the clustering has Min-Sp $\alpha + 1$ then all points in $\mathcal{X}_i$, for each $i$, must be in the same group. Moreover, due to the restriction that every $|\mathcal{X}_i| = s_i \in (L/4, L/2)$, we should have exactly 3 $\mathcal{X}_i$'s in each of the $m$ groups. Thus, the answer is 'YES'. $\qquad \square$

## C.3   Proof of Theorem 4.7

The proof is very similar to that of Theorem 4.2.

*Proof.* We make a reduction from the $(T/4, T/2)$-restricted 3-PARTITION problem. Given a multiset $S = \{s_1, \ldots, s_{3m}\}$ of positive integers that satisfies $\sum_i s_i = mT$, the 3-PARTITION problem consists of deciding whether or not there exists a partition of $S$ into $m$ triples such that the sum of the numbers in each one is equal to $T$. In the $(T/4, T/2)$-restricted 3-PARTITION problem, there is an additional requirement that each number of $S$ should be in the interval $(T/4, T/2)$. This problem is strongly NP-COMPLETE Garey & Johnson (1979)

The instance $I = (X, k, L, \texttt{dist})$ for our clustering problem is built as follows: we set $L = T$, $k = m$; for $i = 1, \ldots, 3m$ let $\mathcal{X}_i$ be a set with $s_i$ points so that the distance between points in the same group $\mathcal{X}_i$ is 1/2 while the distance between points in different groups is $\alpha$. We set $\mathcal{X} = \mathcal{X}_1 \cup \ldots \cup \mathcal{X}_{3m}$.

Let us assume that there is an $(1, \frac{k-2}{k-1} + \frac{1}{\alpha(k-1)})$-approximation for our problem and let $\mathcal{C}$ be the clustering returned by this algorithm for instance $I$. We argue that the answer to the 3-PARTITION problem is 'YES' if and only if MST-Sp$(\mathcal{C}) = (k-1)\alpha$.

First, we show that if the answer is 'YES', there is a $k$-clustering $\mathcal{C}^*$ for $I$ with MST-Sp$(\mathcal{C}^*) = (k-1)\alpha$. In fact, let $S_1, \ldots, S_m$ be a solution of the 3-PARTITION problem and let $\{s_{i_1}, s_{i_2}, s_{i_3}\}$ the numbers in $S_i$. Let $\mathcal{C}^*$ be a $k$-clustering where the $i$th group is comprised by all points in $\mathcal{X}_{i_1} \cup \mathcal{X}_{i_2} \cup \mathcal{X}_{i_3}$. Clearly, each group has $L$ points and the MST-Sp of this clustering is $(k-1)\alpha$. Since our algorithm is a $(1, \frac{k-2}{k-1} + \frac{1}{\alpha(k-1)})$-approximation it returns a clustering $\mathcal{C}$ with MST-Sp$(\mathcal{C})$ $> (k-2)\alpha + 1$. Since the MST-Sp of any clustering for instance $I$ is either $(k-1)\alpha$ or at most $(k-2)\alpha + 1/2$ we have that MST-Sp$(\mathcal{C}) = (k-1)\alpha$

Table 3: Comparison of `MST-Sp` of Algorithm 2 with the upper bound given by Lemma 4.4

| | k | MST-Sp | $\sum_{\ell=2}^{k}$ Min-Sp$(\mathcal{A}'_\ell)$ | Approximation Ratio | $1/H_{k-1}$ |
|---|---|---|---|---|---|
| anuran | 10 | 1.87 | 2.42 | 0.77 | 0.35 |
| avila | 12 | 0.81 | 1.48 | 0.55 | 0.34 |
| collins | 30 | 12.42 | 13.81 | 0.9 | 0.33 |
| digits | 10 | 178.22 | 201.47 | 0.88 | 0.32 |
| letter | 26 | 5.67 | 5.76 | 0.98 | 0.31 |
| mice | 8 | 5.66 | 7.12 | 0.79 | 0.31 |
| newsgroups | 20 | 19 | 19 | 1 | 0.3 |
| pendigits | 10 | 217.01 | 303.37 | 0.72 | 0.3 |
| sensorless | 11 | 1.36 | 2.23 | 0.61 | 0.29 |
| vowel | 11 | 4.94 | 5.57 | 0.89 | 0.29 |

On the other hand, if the clustering has Min-Sp $(k-1)\alpha$ then all points in $\mathcal{X}_i$, for each $i$, must be in the same group. Moreover, due to the restriction that every $|\mathcal{X}_i| = s_i \in (L/4, L/2)$, we should have exactly 3 $\mathcal{X}_i$'s in each of the $m$ groups. Thus, the answer is 'YES'.

$\square$

# D   Experiments: Additional Information

Experiments were run in an Ubuntu 20.04.5 LTS with 40 cores and 115 GB RAM. The repository of the project at `https://anonymous.4open.science/r/SizeConstrainedSpacing-B260` contains the code and instructions needed to generate the experimental data analyzed in the paper.

## D.1   `MST-Sp`: comparison between empirical results and upper bound of Algorithm 2

As mentioned in Section 5, Algorithm 2 can in practice obtain clusterings that are much closer to the optimal `MST-Sp` than the prediction guaranteed by Theorem 4.6. In Table 3, we present, for each dataset: the average `MST-Sp` obtained by Algorithm 2; the upper bound on the `MST-Sp` given by the sum of the `Min-Sp` for all partitions found by an execution of the algorithm; the approximation ratio of Algorithm 2, given by its `MST-Sp` divided by the upper bound; and the theoretical approximation ratio $1/H_{k-1}$ from Theorem 4.6.

For all 10 datasets, Algorithm 2 performs significantly better than its theoretical approximation ratio. The smallest gap between theoretical and empirical result occurs for `avila` dataset, in which the algorithm is 21 percentage points closer to the optimal `MST-Sp` than Theorem 4.6 guarantees; On the other extreme, for dataset `newsgroups`, it actually achieves the best possible `MST-Sp`. These results increase our confidence that Algorithm 2 is a good option for finding separated groups.

## D.2   Average size of smallest clusters

Table 4 presents the average size of the smallest group generated by Algorithms 1 and 2 and $k$-means. Values tend to be close across all algorithms, and for all iterations of the experiments the smallest group returned by Algorithms 1 and 2 is at least as large as the smallest group from the corresponding $k$-means clustering; recall that $L$ is set as $4s/3$, where $s$ is the size of the smallest group produced by $k$-means. In particular, thanks to the rule of iterating from the largest cluster to the smallest when building our $k$-clustering from an $\ell$-clustering, the theoretical possibility that the smallest group induced by Algorithm 2 is $1/2$ of the desired size does not appear to happen in practice.

## D.3   Fast version of Algorithm 2

As mentioned in Section 5, the $\log k$ bound of Theorem 4.6 is still valid for Algorithm 2 if, instead of investigating all values of $\ell$ from 2 to $k$, it considers only the values that can be written as $\lceil k/2^t \rceil$, for $t = 0, \ldots, \lfloor \log k \rfloor$. In Table 5 we compare the results of this fast version of the algorithm with those of the full version.

Table 4: Average size of the smallest cluster in a $k$-clustering, per algorithm and dataset.

| | Dimensions | | Average size of smallest cluster | | |
|---|---|---|---|---|---|
| | n | k | Algorithm 1 | Algorithm 2 | $k$-means |
| anuran | 7,195 | 10 | 264.5 | 266.7 | 264.1 |
| avila | 20,867 | 12 | 85.3 | 85.4 | 83.2 |
| collins | 1,000 | 30 | 7.7 | 7.7 | 7.7 |
| digits | 1,797 | 10 | 96.2 | 96.2 | 92.7 |
| letter | 20,000 | 26 | 192 | 258.7 | 191.5 |
| mice | 552 | 8 | 47.6 | 47.6 | 47 |
| newsgroups | 18,846 | 20 | 188.1 | 188.2 | 188.1 |
| pendigits | 10,992 | 10 | 483.2 | 476.4 | 463.8 |
| sensorless | 58,509 | 11 | 1,842.5 | 1,725.5 | 1,655.6 |
| vowel | 990 | 11 | 57.9 | 57.9 | 57.9 |

Table 5: `Min-Sp`, `MST-Sp` and execution time for Algorithm 2.

| | Dimensions | | MST-Sp | | Time (seconds) | |
|---|---|---|---|---|---|---|
| | n | k | Fast | Full | Fast | Full |
| anuran | 7,195 | 10 | 1.71 | **1.87** | **2.55** | 6.77 |
| avila | 20,867 | 12 | 0.77 | **0.81** | **20.47** | 62.51 |
| collins | 1,000 | 30 | **12.42** | **12.42** | **0.76** | 4.48 |
| digits | 1,797 | 10 | **178.22** | **178.22** | **0.55** | 1.78 |
| letter | 20,000 | 26 | 5.66 | **5.67** | **23.44** | 116.44 |
| mice | 552 | 8 | **5.66** | **5.66** | **0.27** | 0.44 |
| newsgroups | 18,846 | 20 | **19** | **19** | 307.58 | 440.38 |
| pendigits | 10,992 | 10 | 215.11 | **217.01** | **6.13** | 16.02 |
| sensorless | 58,509 | 11 | 1.34 | **1.36** | **274.00** | 701.42 |
| vowel | 990 | 11 | **4.94** | **4.94** | **0.18** | 0.59 |

Even considering the overhead of running the single-linkage algorithm, which cannot be avoided for both versions of Algorithm 2, we see a reduction of at least 30% in the algorithm's running time when using the fast version. The loss in terms of `MST-Sp`, on the other hand, is less than 10% in the worst scenario, and in 5 of the 10 datasets analyzed both versions return the same clustering.

### D.4 Distribution of results for `Min-Sp` and `MST-Sp`

Figures 3 and 4 show the boxplots for the `Min-Sp` and the `MST-Sp`, respectively, per dataset and algorithm. Algorithm 2 presents some large variations, when compared to both k-means and Algorithm 1, in terms of `Min-Sp` (Figure 3) for some datasets; as it is designed to maximize the `MST-Sp`, this behavior in the other metric presented here should not be too concerning. Also in terms of `Min-Sp`, both algorithms presented in the paper clearly outperform k-means in almost all datasets, even considering the variation in results. The same can be said for the `MST-Sp` (Figure 4), in which, additionally, the range of results returned by Algorithm 2 is much more in line with those returned by the two other algorithms.

### D.5 Trade-off between size of smallest cluster and inter-group separability criteria

In Figure 5 we present scatterplots for our 5 smallest datasets showing how the quality of the clusterings generated by Algorithms 1 and 2 (considering, respectively, the `Min-Sp` and the `MST-Sp` as criteria) increases as we allow for clusters of smaller sizes. For all datasets, as expected, allowing for smallest clusters leads to higher `Min-Sp` and `MST-Sp`. It is still noteworthy that the algorithms presented in this paper can be used not only to find a good partition with a hard limit on the size of the smallest cluster, but also to find the best balance between minimum size and a good separation of clusters.

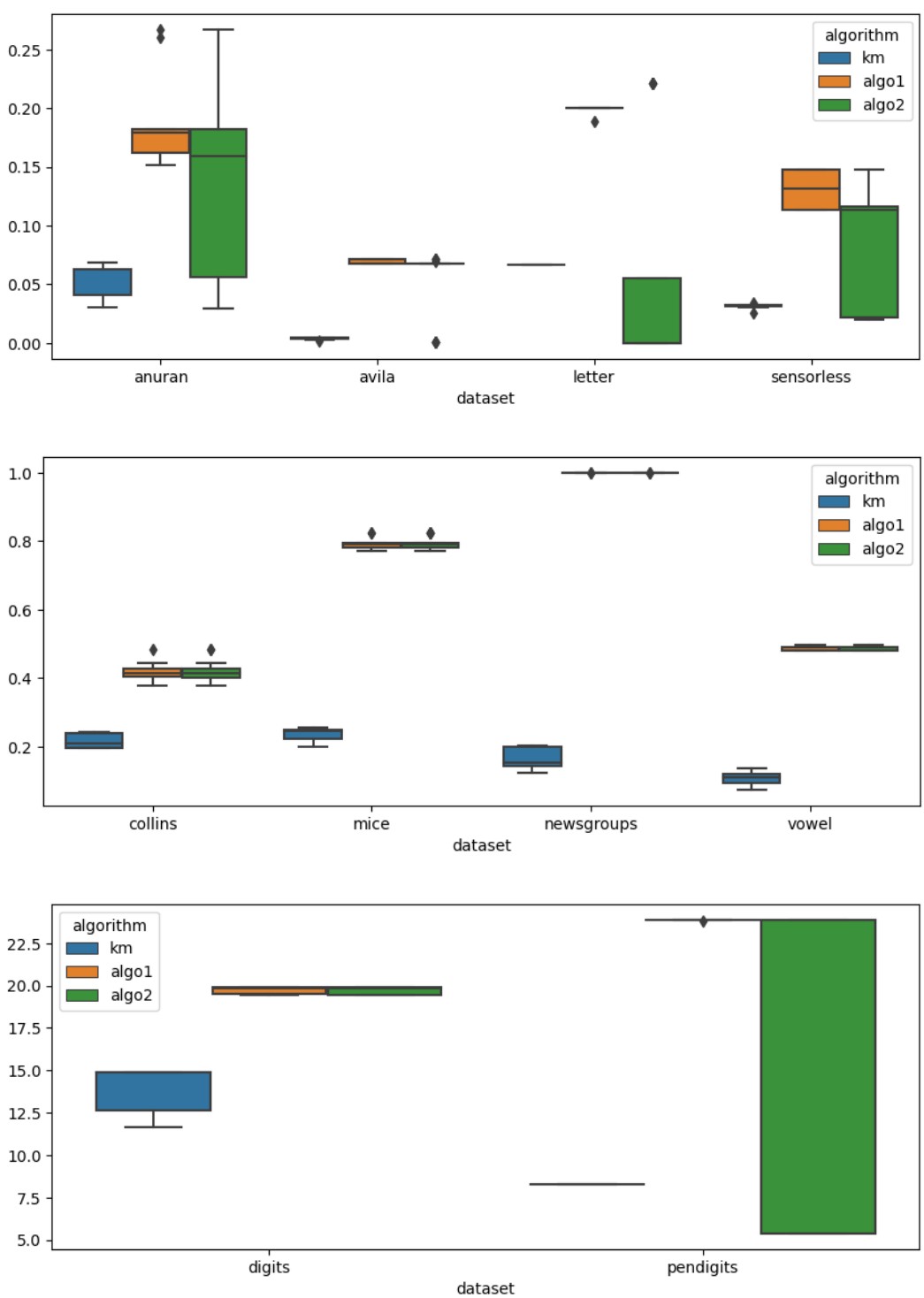

Figure 3: Boxplots of the `Min-Sp` per dataset and algorithm.

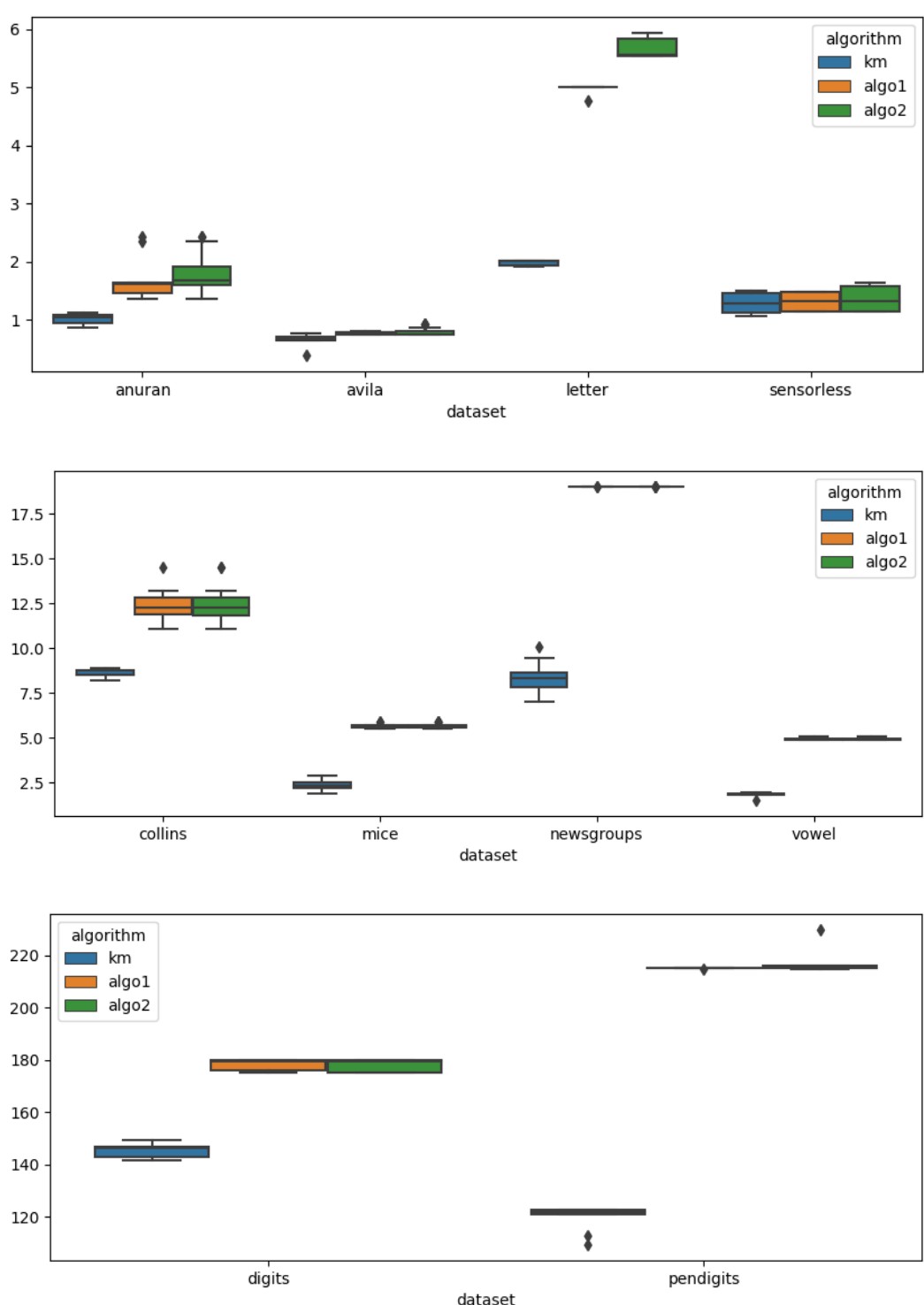

Figure 4: Boxplots of the `MST-Sp` per dataset and algorithm.

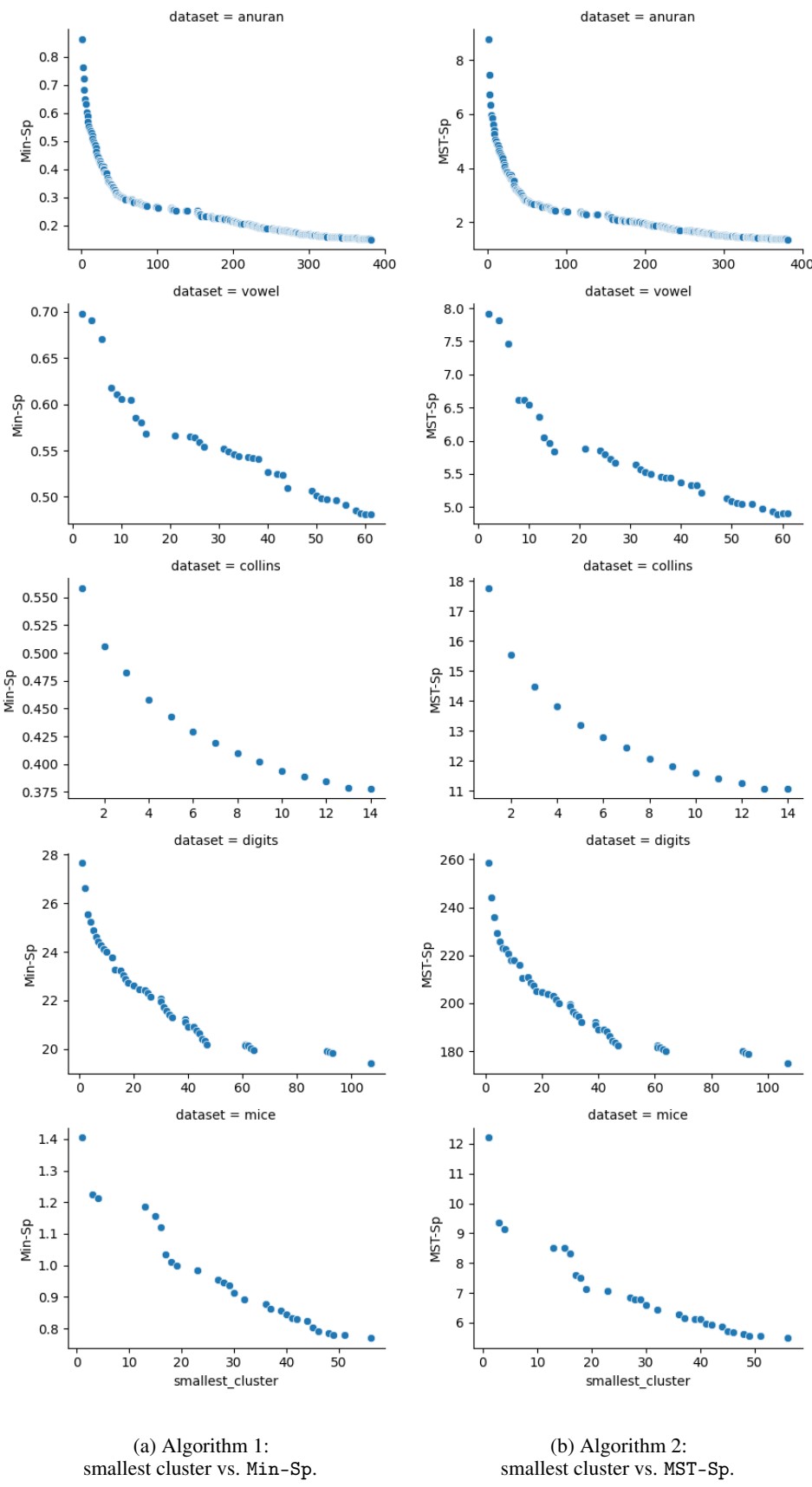

(a) Algorithm 1:
smallest cluster vs. `Min-Sp`.

(b) Algorithm 2:
smallest cluster vs. `MST-Sp`.

Figure 5: Trade-off between the size of the smallest cluster and the separability criteria.

Table 6: Maximum standard deviation of `MST-Sp` for Algorithm 2.

|  | Dimensions | | MST-Sp | |
|---|---|---|---|---|
|  | n | k | $\mu$ | $\max \sigma$ |
| anuran | 7,195 | 10 | 1.87 | - |
| avila | 20,867 | 12 | 0.81 | - |
| collins | 1,000 | 30 | 12.42 | - |
| digits | 1,797 | 10 | 178.22 | - |
| letter | 20,000 | 26 | 5.67 | 0.34 |
| mice | 552 | 8 | 5.66 | - |
| newsgroups | 18,846 | 20 | 19 | - |
| pendigits | 10,992 | 10 | 217.01 | - |
| sensorless | 58,509 | 11 | 1.96 | 0.002 |
| vowel | 990 | 11 | 4.94 | - |

Table 7: Quadratic loss ($k$-means cost) for Algorithms 1 and 2.

|  | Dimensions | | Loss (relative to $k$-means) | |
|---|---|---|---|---|
|  | n | k | Algorithm 1 | Algorithm 2 |
| anuran | 7,195 | 10 | 2.50 | **2.07** |
| avila | 20,867 | 12 | **2.81** | **2.81** |
| collins | 1,000 | 30 | **2.33** | **2.33** |
| digits | 1,797 | 10 | **1.33** | **1.33** |
| letter | 20,000 | 26 | **2.30** | 2.52 |
| mice | 552 | 8 | **2.52** | **2.52** |
| newsgroups | 18,846 | 20 | **1.05** | **1.05** |
| pendigits | 10,992 | 10 | 2.18 | **2.10** |
| sensorless | 58,509 | 11 | **4.32** | 5.25 |
| vowel | 990 | 11 | **2.07** | **2.07** |

### D.6 Effect of randomness on Algorithm 2's results

While Algorithm 1 is fully deterministic, in Algorithm 2 the split of clusters from an $\ell$-clustering to turn it into a $k$-clustering is performed randomly. In practice, however, this does not affect the results of the algorithm.

For each dataset, we ran 10 seeded iterations of Algorithm 2 for each value of $L$ used in the experiments. We then calculate the standard deviation of the `MST-Sp` for each value of $L$. As shown in Table 6, for 8 of the datasets analyzed the `MST-Sp` of the clustering returned by Algorithm 2 is always the same for a given value of $L$; for `letter` and `sensorless`, there is some variation, but it is very small compared to the average `MST-Sp` returned by the algorithm.

### D.7 Relative quadratic loss for Algorithms 1 and 2

In Table 7 we present the quadratic loss of both Algorithm 1 and Algorithm 2 as a multiple of the loss incurred by the $k$-means algorithm, which is specifically designed to minimize this loss. As expected, since both algorithms were devised for maximizing inter-group criteria, they perform poorly in light of this intra-group loss function — with the sole exception of the `newsgroups` dataset, for which both algorithms incur a loss only 5% above that of $k$-means. Across datasets, the performance of both algorithms is similar for this loss, with only small variations.

