# OpenReview forum: "Optimization of Inter-group criteria for clustering with minimum size constraints"
_NeurIPS.cc/2023/Conference — NeurIPS 2023 poster_

### Official Review · Reviewer_GUvh · 2023-06-09

**Soundness:** 4 excellent
**Presentation:** 3 good
**Contribution:** 3 good
**Rating:** 8
**Confidence:** 4

**Summary:**

The authors of this paper consider metric clustering problems, where the goal is to maximize inter-clustering objectives. Such objectives measure how "distinct" different clusters are. The first objective they consider is the minimum spacing one (MS). According to this, the goal is to maximize the minimum distance between any resulting clusters, where the minimum distance between clusters is naturally defined as the minimum distance between any of their points. The second objective they consider is maximizing the minimum spanning tree (MST) distance. Here, the MST distance is defined on the complete graph representing the resulting clusters. I would say that the latter objective captures a more general/global notion of connectivity/similarity compared to the earlier one.

At first, the authors state that the single-linkage algorithm by Kleinberg and Tardos maximizes the MS objective, and this is something already known in the literature. However, they also show that this algorithm satisfies MST as well. More interestingly, they show that any algorithm satisfying MST will also satisfy MS.

Moving forward, the authors tackle a constrained version of the problem, where besides minimizing the appropriate objective, they want to create clusters of size at least L, for some given L. For the constrained versions of the their problems, they give their most algorithmically involved results, which consist of bicriteria approximation algorithms followed by corresponding lower bounds.

**Strengths:**

1) The paper is technically solid and well-written.

2) The inter-clustering objective is cute and quite novel. From the theoretical perspective it opens up a new direction for clustering research.

3) The results are pretty tight. The characterization of Theorem 3.3 is solid and impressive and the lower bound algorithms are accompanied by the proper inapproximability results.

**Weaknesses:**

1) From a practical motivational perspective, I am not convinced about the use-cases of such a model. The authors should have elaborated more there, giving more fitting applications. This is honestly my only concern for this paper. I can easily see this paper getting into a pure algorithms conference like APPROX-RANDON or ICALP, but I feel like for NeurIPS standards there should have been a more detailed discussion on how this inter-group model (with the objective functions considered) could fly in real-life scenarios (perhaps by giving concrete specific use-cases). Also, from a practical perspective I can't see the necessity of the lower bound constraint.

If it wasn't for the lack of motivation my score would most likely be higher.

**Questions:**

1) I think that the following works are a bit related to yours. Under the broader umbrella of inter-group clustering. The goal in those papers is to make sure that on average, each point is closer to points in its own cluster compared to other clusters. To me this feels like an inter-group objective.

https://arxiv.org/abs/2006.04960
https://arxiv.org/abs/2207.03600

2) In Figure 1 where you say that the left clustering is not natural, it is not really clear what you mean. I suppose that you want to show that solving MS can negatively impact MST.

---

> ### Author Rebuttal · Authors · 2023-08-07
>
> We are quite happy that you appreciated our work!
>
> In our comment directed to all referees, we discuss the fit for Neurips and potential applications.
>
> Questions:
>
> 1. About the papers you mentioned
>
>     **Reply.** Thanks for pointing out the papers! Indeed, they are related to ours and we will cite them in our new version.
>
> 2. In Figure 1 where you say that the left clustering is not natural, it is not really clear what you mean. I suppose that you want to show that solving MS can negatively impact MST.
>
>     **Reply.** The main point of the figure is that saying that single-linkage maximizes the minimum spacing may not fully describe its behavior, as other partitions that do the same may look very distinct from that generated by single-linkage. We will try to make this clearer in the revised version of the paper.

---

> > ### Comment · Reviewer_GUvh · 2023-08-11
> > **Adjustment of my score**
> >
> > I have read all reviews and all rebuttals. Based on that I decided to update my score to an 8 (Strong Accept). Here are the reasons for this adjustment:
> >
> >    1.  My only concern about the paper when I first read it, was the lack of motivation. I am satisfied with the example that the authors gave in their rebuttal and I encourage them to include it in the main paper.
> >
> >    2. I do not agree with reviewer FsJN saying that this paper/problem is of no interest for the NeurIPS community. I can honestly attach a 50+ list of papers on clustering from NeurIPS/ICML/AAAI/AISTATS/etc that are of a more algorithmic/theoretical flavor, yet they have been published in these venues, have been accepted by the community and some of them have even sparked new and exciting research directions (the seminal "Fair Clustering Through Fairlets" is the first example that comes to mind).
> >
> >    3. I do not agree with reviewer FmRy saying that the results presented in the paper are trivial. For one thing the authors did a great job refuting the claim for the two "simpler" solutions the reviewer proposed. In addition, even though the algorithms are relatively simple and use some known techniques (clearly every clustering algorithm that has a minmax distance objective will first guess the optimal value), in my opinion there's ingenuity in the proofs.
> >
> > Overall, I believe that it would be a pity if this paper is not accepted at NeurIPS.

---

> > > ### Author Response · Authors · 2023-08-11
> > >
> > > Thanks for engaging in the discussion!
> > >
> > > We greatly appreciate your kind words regarding our submission.
> > >
> > > We will definitely expand the discussion about motivations and applications in the revised version of the paper.

---

### Official Review · Reviewer_FsJN · 2023-06-29

**Soundness:** 3 good
**Presentation:** 3 good
**Contribution:** 2 fair
**Rating:** 3
**Confidence:** 4

**Summary:**

This paper considers the problem where we want to cluster $n$ points into $k$ clusters with the clustering objective trying to maximize the minimum distances between clusters (either the actual minimum distance or the weight of an MST defined between clusters). The authors show that greedy agglomerative clustering (stopping when $k$ clusters are formed) with single-linkage optimizes both objectives. The authors also consider the scenario where the user specifies a minimum cluster size as an additional constraint. Approximation algorithms for this setting are presented and hardness of approximation results are shown. Empirical results are shown on publicly available datasets.

**Strengths:**

* The paper is well-written
* The problem is interesting from a theoretical perspective
* The theoretical results are sound, albeit not surprising

**Weaknesses:**

* The motivation for the problem setting is very weak. It is unclear where this would be used in practice.
* The results presented are probably not of high interest to the NeurIPS community. The theoretical results are solid, but not surprising, and the applications of this method are not presented. Could this be more appropriate for a theory venue?
* The experimental results do not convey much about the performance of the algorithms except for their relative performance, which is not very dramatic. Are there other experiments that could be run to show why someone would want to use Algorithm 1 or 2 over another approach?

**Questions:**

* What is another example application other than the diversity example given on lines 34-38?
* What benefits do Min-Sp and MST-Sp have over other clustering objectives? Why would a user choose this objective over another objective?
* How might one select the minimum cluster size? Could this be an issue just as difficult as selecting the number of clusters $k$?

**Limitations:**

Limitations are adequately addressed.

---

> ### Author Rebuttal · Authors · 2023-08-07
>
> Thanks again for your time and effort!
>
> We discuss applications and the fit for Neurips in our comments addressed to all reviewers.
>
> Questions:
>
> *The experimental results do not convey much about the performance of the algorithms except for their relative performance, which is not very dramatic. Are there other experiments that could be run to show why someone would want to use Algorithm 1 or 2 over another approach?*
>
> **Reply:** Apparently, you are only taking into account the experimental results presented in the body (main submission). Is this the case?
>
> By taking into account also the supplemental material, we feel that our experiments do more than show the algorithms’ relative performance:
>
> 1. In Table 1, we show that our algorithms clearly outperform $k$-means for the separability metrics analyzed in the paper. We believe this is what you are referring to when talking about relative performance.
>
> 2. In Table 3, we compare the results of Algorithm 2 with the upper bound of MST-Sp given by Lemma 4.4, showing that the results are frequently much closer to the optimal than the approximation guaranteed by Theorem 4.6. That is, the absolute performance of this algorithm is also good (close to the optimal in many cases).
>
> 3. In Table 5, we present the results of a different, faster version of Algorithm 2, which are quite close to, and frequently the same as, the results of the algorithm’s full version. That is, we can achieve good results (in both relative and absolute terms) quite fast if necessary.
>
> 4. In Figures 3 and 4, we show that the results of Algorithms 1 and 2 for their respective separability metrics are robust, in two senses:
> Their results do not vary much across iterations.
> Their results are clearly superior to $k$-means, even considering possible variations across iterations.
>
> 5. In Figure 5, we show the trade-off between the minimum cluster size and the separability criteria. This can be important when trying to establish the optimal minimum cluster size (see also our answer to your question below).
>
>
> *What is another example application other than the diversity example given on lines 34-38?*
>
> **Reply:**  See our comment directed to all referees.
>
> *What benefits do Min-Sp and MST-Sp have over other clustering objectives? Why would a user choose this objective over another objective?*
>
> **Reply:** One may choose these objectives to generate groups that are far from each other because these objectives are natural (easy to understand) and they have algorithms with provable guarantees that are fairly simple to implement (the ones we proposed here).
>
> *How might one select the minimum cluster size? Could this be an issue just as difficult as selecting the number of clusters?*
>
> **Reply:** We view the minimum cluster size as an issue that depends largely on the specifics of each use case.  We present in Figure 5 of the supplementary material plots representing the trade-off between minimum cluster size and the separability criteria. Analyzing this trade-off may be very useful when deciding on the minimum cluster size; as we mention in section D.5,  “the algorithms presented in this paper can be used not only to find a good partition with a hard limit on the size of the smallest cluster, but also to find the best balance between minimum size and a good separation of clusters.”

---

> > ### Comment · Reviewer_FsJN · 2023-08-14
> >
> > Thank you for your response. The content of the main paper should be enough to convince a reader of the efficacy of the proposed approach. The supplemental material is just that, *supplemental*.  Authors cannot skirt the page limit set by saying something is in the supplemental material. The empirical results in the main paper are still not terribly convincing in there own right, but do show some minor claims argued in the paper. That being said, I am willing to increase my score if some of the more convincing empirical results are added to the main paper. I think that there is plenty of space if some of the proofs in the main paper are moved to the supplemental material.
> >
> > My concerns about motivation have been mitigated by the additional potential applications provided in the global comment. Thank you for providing this detailed response.

---

> > > ### Author Response · Authors · 2023-08-14
> > >
> > > Thanks for replying to our rebuttal and it is good to read that you are willing to increase your score!
> > >
> > > We agree with you that the  “main paper should be enough to convince a reader of the efficacy of the proposed approach”. Given the space constraint, we had to make some choices between presenting more proofs or more experimental results and, unfortunately, on some occasions, our choices are not the best possible.
> > >
> > > For the final version, we will:
> > >
> > > 1. Expand the discussion about potential applications in the introduction
> > >
> > > 2. Move the most relevant empirical results from the Supp. Material to the main paper.
> > >
> > > 3. Add more intuition to the approximation achieved by Algorithm 2.
> > >
> > > This will be possible by moving some of the proofs to the appendix, as you suggested (prioritizing the proofs of the Lemmas in Section 4.2), and by also taking advantage of an extra content page that is given to the accepted papers.

---

### Official Review · Reviewer_FmRy · 2023-07-06

**Soundness:** 2 fair
**Presentation:** 2 fair
**Contribution:** 2 fair
**Rating:** 5
**Confidence:** 4

**Summary:**

The paper considers the following problem: given a metric space on a set of $n$ points and a parameter $k$, partition the points into $k$ clusters $C_1, \ldots, C_k$. The paper considers two objective functions: consider a metric  with vertices $X$ as the set of clusters and distance between two clusters being the minimum distance between two points belonging to these clusters respectively. The first objective is to maximize the minimum distance between any two points in $X$, and the second objective is to maximize the cost of the min. spanning tree on $X$. It is easy to show that the greedy algorithm for MST can be modified to solve these two problems in polynomial time (essentially run the Kruskal's algorithm till only $k$ components remain).

However, the authors observe that this can lead to skewed clusters with some clusters being too small. So they propose the following problem: consider the same settings as above, but now each cluster must have at least $L$ points, where $L$ is another parameter provided in the input. They give the following polynomial time guarantees for the two problems: (i) for the max min. cluster separation objective, they give an algorithm of optimal value but each cluster has size at least $L(1-\epsilon)$, where $\epsilon > 0$ is any constant, and (ii) for the MST objective, they give an algorithm of objective value at least $1/\log k$ times the optimal value, and each cluster size is at least $L(1-\epsilon)$. They show that  modifications of the greedy heuristic along with known results for a schedling problem work for the first problem, whereas the second one requires more details.

**Strengths:**

1. Introduce an interesting problem.
2. Give near-tight bounds for the max min cluster separation problem, and leave an interesting open problem for the max min spanning tree problem.

**Weaknesses:**

It seems to me that the two problems can be solved in a simpler manner without appealing to the greedy heuristic. Assume that the metric on the $n$ points is given by a weighted graph $G$.

1. First consider the max min cluster separation problem: suppose we know the optimal value $T$ (which we can guess by binary search). Now remove all edges of length less than $T$ from the $G$:  these edges cannot appear between two distinct clusters. Now consider the components formed -- they should be more than $k$. Now merge these components using the scheduling problem used in the paper.

2. For the max. mst problem: again suppose we know the optimal value, say $T$. Now consider the optimal tree on the $k$ vertices $X$ -- call this tree $F$. Ignore the edges of cost less than $T/2k$ from $F$ -- the total cost of such edges is at most $T/2$. Now consider the edges in $F$ whose cost lies in the range $[T/2k, T]$. Divide this range into $\log k$ buckets of exponentially increasing range. So there is an index $i$ such that the edges in $F$ (call these edges $E_i$) with cost in the range $[\frac{T}{2^{i+1} k}, \frac{T}{2^i k}]$ have total cost at least $\frac{T}{2 \log k}$. Now remove all edges of length at least $\frac{T}{2^{i+1} k}$ from $G$. It is not difficult to show that if $E_i$ has $a$ edges, then removing such edges from $G$ will create at least $a$ components. Now we form clusters so that no cluster contains points from two distinct components.

**Questions:**

1. It is not clear if the problems are non-trivial or follow from known ideas/techniques.

2. Please explain if instead of placing lower bound of $L$, placing an upper bound of $L$ will change the problem significantly.

**Limitations:**

No negative impact.

---

> ### Author Rebuttal · Authors · 2023-08-07
>
> We thank the reviewer!
>
> However, we are a bit surprised by his/her evaluation, as discussed below.
>
> **Score**
>
> It is not clear what are the soundness and presentation problems in our work that justify a score 2 for these dimensions; there is no comment about it.
>
> **Weakness**
>
> The weakness that the referee identified in our work is that “the two problems can be solved in a simpler manner without appealing to the greedy heuristic”.
>
> We do not see why the solutions that the referee proposed are simpler than ours. Even if they are, we believe that our solutions are simple enough, as we managed to implement them without any major obstacles.
>
> More importantly, as far as we understand, the solution proposed for the min-sp is essentially proposed in the paper while the one for the minimum MST is not detailed enough, which makes it hard to verify its correctness and how simple it is. We explain below:
>
> 1. The solution proposed for the maximization of min-sp is very similar to the one we mention in lines 197-199. The only differences that we identify are:
>
>     - instead of adding edges (Kruskal), the proposed solution removes them.
>
>     - the proposed binary search does not run in strongly polynomial time (due to the logarithm on the maximum possible cost) while ours does.
>
> 2. Important details are missing from the solution for the MST-sp:
>     - the reviewer mentions an index $i$ that will be used to "remove all edges of length at least $\frac{T}{2^{i+1}k}$ from $G$". How can we find this index without having the tree $F$? Shall we try all the possibilities for $i$?
>
>     - The last step of this solution is to "form clusters so that no cluster contains points from two distinct components." It is not clear how this step shall be implemented. **More seriously**, we agree that at least $a$ components are created when the edges are removed from $G$, but it may happen that more than $k$ components can be created (some of the $k$ clusters of $F$ may be split into many components); in this case, the proposed rule (“no cluster contains points from distinct components”) will not lead to the production of $k$ clusters. Wouldn't this be a problem in the proposed solution?
>
> **Questions:**
>
> *It is not clear if the problems are non-trivial or follow from known ideas/techniques.*
>
> **Reply:** We put considerable energy into finding meaningful and tractable formulations and also coming up with approximation algorithms and inapproximability results. The relatively simple proofs presented in the paper - in particular, those from Section 3 that the reviewer classifies as “easy” in his/her summary - are the result of successive refinement; the first versions were much messier and more intricate than the current ones. Many researchers see simplicity as a virtue rather than a weakness.
>
> *Please explain if instead of placing lower bound of $L$, placing an upper bound of $L$
>  	will change the problem significantly.*
>
> **Reply:** The techniques we used do not seem to apply directly when we use upper bound constraints.
>
> In light of the above discussion, we kindly request that the reviewer revise his/her scores.

---

> > ### Comment · Reviewer_FmRy · 2023-08-11
> >
> > Thanks for replying.
> >
> > 1. For the first problem, what I meant was that the algorithm follows from known ideas: guessing optimal value, removing edges of high cost (e,g,. as in bottlneck spanning trees) are known ideas. So the algorithm description looked more complex than it should be. Also guessing OPT can be done in strongly polynomial time: OPT can only take value from one of the pair-wise distances, so there are only $n^2$ choices for OPT. We can do a binary search on this. Again, this is a standard idea.
> >
> > 2. For the second problem, I agree with your objection, this is a point I had missed. Thanks for correcting me. One could try to fix it, but then it would take one far from the comment I had made. So I think your algoithm is non-trivial here.  The presentation here can be improved by first explaining the intuition behing the $O(\log k)$ ratio in the approximation -- exactly where it is coming from in the analysis.

---

> > > ### Author Response · Authors · 2023-08-11
> > >
> > > Thanks for engaging in the discussion!
> > >
> > > We are happy with your comment regarding Algorithm 2: *“now I think that our is non-trivial”*. We will add more intuition to the revised version of the paper.
> > >
> > > We would like to bring to your attention one subtle issue regarding the presentation of Algorithm 1.
> > >
> > > We believe that your proposal to guess OPT and then do a binary search may lead the reader to think there is monotonicity in the values ​​of minimum spacing for the clustering produced by the algorithm.
> > > However, since we are using an approximation algorithm for the scheduling step this monotonicity does not necessarily exist.
> > >
> > > The monotonicity in the values of minimum spacing does exist for the clustering obtained before running the scheduling step (right after running Kruskar or removing edges as you propose). We believe the way it is presented in the paper is more clear in this sense.

---

> > > > ### Comment · Reviewer_FmRy · 2023-08-20
> > > >
> > > > I have updated my score to "borderline accept" given that the second algorithm is more non-trivial. However I am still not convicned that the first algortuhm has novelty in it. Also, I think presentation of the algorithms can be improved.

---

### Official Review · Reviewer_Z252 · 2023-07-07

**Soundness:** 4 excellent
**Presentation:** 3 good
**Contribution:** 3 good
**Rating:** 7
**Confidence:** 4

**Summary:**

The paper presents a body of theoretical results addressing the problems of minimum-spacing and minimum spanning tree clustering; both based on the concept of finding clustering solutions which maximise the inter-cluster distance(s). The former maximises the distance between the pair of closest clusters while the latter maximises the total distances between clusters along a tree connecting them (where set distances are defined as the minimum pairwise distances between elements in different sets).

The single-linkage agglomerative method is known to maximise inter-cluster distances but is also known to have poor properties empirically (notably the chaining effect and isolation of "outliers" and "noise"  over the construction of entire clusters). Based on this the authors propose imposing size-constraints which bound the size of the smallest cluster.

The authors show that hard constraints of this type for the two objectives studied are APX-hard, and so propose relaxations which allow a (1-\epsilon) factor on the minimum cluster size but admit polynomial time algorithms for the minimum spacing, and in the case of the minimum spanning tree relaxations which also involve the average cluster size relative to the "hard constraint size" (being a more substantial relaxation the more balanced the cluster sizes need to be) as well as an approximation factor on the objective function given by the inverse of the k-1 th harmonic number (k being the number of clusters).

The paper concludes with some experiments documenting the performance of their proposed algorithms.

**Strengths:**

The work is extremely thorough and addresses an important problem, which is that maximum inter-cluster distances are very intuitive as objectives for clustering but have poor robustness. The size constraint proposed is natural and intuitive, and the authors have clearly done more than their due diligence in studying the extending the problem.

The paper is also, subject to some technical knowledge, clear and easy to follow.

**Weaknesses:**

Nothing substantial. Arguably some missed pieces of literature:
- Early work on MST clustering: Zahn, Charles T. "Graph-theoretical methods for detecting and describing gestalt clusters." IEEE Transactions on computers 100.1 (1971): 68-86.
- Connection between Min-Sp and spectral clustering: Hofmeyr, David P. "Connecting spectral clustering to maximum margins and level sets." The Journal of Machine Learning Research 21.1 (2020): 630-664.
- Maximum Margin Clustering: Xu, Linli, James Neufeld, Bryce Larson, and Dale Schuurmans. "Maximum margin clustering." Advances in neural information processing systems 17 (2004).

Typos and similar:
- Line 111: plural of vertex is "vertices"
- Lines 126-7: "the the cut property... and the cut property"???
- Include the explicit version of PTAS

**Questions:**

None which I can think of. Good work

**Limitations:**

None noted

---

> ### Author Rebuttal · Authors · 2023-08-07
>
> We greatly appreciate your kind words! We will be sure to review our text for typos and to incorporate the papers you mentioned to our Related Work section.

---

### Author Rebuttal · Authors · 2023-08-07

**To all Reviewers**

We thank all the referees for your time and effort. We are happy that some referees classified our work as solid and extremely thorough and that all referees found our problem interesting.

We have specific comments for each referee, but in the current one, we discuss potential applications for our work and the issue of whether it is interesting for the Neurips community,

**Fit to Neurips Community**

 Our paper is essentially dedicated to studying problems that optimize the same functions that are optimized by the Single-Link algorithm, the Min-SP, and the Sum-Sp, with the latter being identified in our work. The Single-Link algorithm is a classical algorithm and it is available in most (if not all) Machine Learning packages. Moreover, it has been the subject of a number of studies that were published in Neurips and JMLR  such as  [Kleinberg, Neurips 2002, cited in the paper],  [Xu, Linli et. al, Neurips 2004, reference 1 below], [Carlsson, G. E. and Mémoli, F, JMLR 2010, cited in the paper], [Hofmeyr, David P, JMLR 2020, reference 2 below].

It is worth mentioning a connection of single-link with the concept of “Individual preference stability” studied in a recent ICML paper [3]. The connection relies on the fact that the single-link construction guarantees that the closest neighbor of each point $p$ is in the same group as $p$.  The motivations for “Individual preference stability” include guaranteeing stability in the sense that no point has the incentive to move from its group to another one and fairness (each point is “happy” with its location).  We thank referee GUvh for pointing out this reference.

In summary, we are contributing to extending the current knowledge regarding an important algorithm in Machine Learning, which, by itself, we believe makes our paper interesting for the Neurips community.

**Potential Applications**

We agree that our discussion on use cases for our algorithms should be expanded, as only a few lines (34-38) were dedicated to presenting a single use case (ensuring diversity in committees).

Here we discuss two other potential use cases: ensuring population diversity in candidate solutions for genetic algorithms [4] and data diversity when training machine learning algorithms [5]. We will add this discussion to the new version of the paper.

When training a machine learning model, ensuring data diversity may be crucial for achieving good results [5]. In situations in which all available data cannot be used for training (e.g. training in the cloud with budget constraints), it is important to have a method for selecting a diverse subset of the data, and our algorithm can be used for this: to select $n = kL$ elements, one can partition the full data set into $k$ clusters, all of them containing at least $L$ members, and then select $L$ elements from each cluster. Using Single-Link to create $kL$ groups and then picking up one element per group is also a possibility but it would increase the probability of an overrepresentation of outliers in the obtained subset, as these outliers would likely be clustered as singletons (see our Figure 2 on the prevalence of singletons when using Single-Link).

Regarding the above application, we note that, as in the case of diverse committees, our algorithm can be used to create not only one but several diverse and disjoint subsets, which might be relevant to generate partitions for cross-validation or for evaluating a model's robustness.  For that,  each subset is obtained by picking exactly one point per group.


For genetic algorithms, maintaining diversity over the iterations is important to ensure a good exploration of the search space [4]. If all candidate solutions become too similar, the algorithm will become too dependent on mutations for improvement, as the offspring of two solutions will likely be similar to its two parents; and mutation alone may not be enough to fully explore the search space. We can apply our algorithm in a similar manner as mentioned above, by partitioning, at each iteration, the solutions into $k$ clusters of minimum size $L$ and selecting the best $L$ solutions from each cluster, according to the objective function of the underlying optimization problem, to maintain the solution population simultaneously optimized and diverse. Using Single-Link could lead to several poor solutions being selected to remain in the population, in case they are clustered as singletons.

We finally note that an algorithm that optimizes intra-group measures (e.g. $k$-means or $k$-medians) would not necessarily guarantee diversity for the aforementioned applications, as points from different groups can be close to each other.

---

### Decision · Program_Chairs · 2023-09-21

**Decision:**

Accept (poster)

**Comment:**

Though there is some variance in the opinions of reviewers, several found that the minimum size constraints formulation here is natural and novel, with a number of valuable contributions in the optimization of inter-group criteria. Moreover, the performance of the method is also promising. As some of the dismissive claims in the negative reviewers are not as clearly supported in my view, my own read tends to agree in finding value that will be of interest to the NeurIPS community.